# Epidemic Diffusion Network of Spain: A Mobility Model to Characterize the Transmission Routes of Disease

**DOI:** 10.3390/ijerph20054356

**Published:** 2023-02-28

**Authors:** Javier Del-Águila-Mejía, David García-García, Ayelén Rojas-Benedicto, Nicolás Rosillo, María Guerrero-Vadillo, Marina Peñuelas, Rebeca Ramis, Diana Gómez-Barroso, Juan de Mata Donado-Campos

**Affiliations:** 1Departamento de Medicina Preventiva y Salud Pública y Microbiología, Facultad de Medicina, Universidad Autónoma de Madrid. C. Arzobispo Morcillo 4, 28029 Madrid, Spain; 2Centro Nacional de Epidemiología, Instituto de Salud Carlos IIII, Calle de Melchor Fernández Almagro 5, 28029 Madrid, Spain; 3Servicio de Medicina Preventiva, Hospital Universitario de Móstoles, Calle Río Júcar s/n, 28935 Móstoles, Spain; 4Consorcio de Investigación Biomédica en Red de Epidemiología y Salud Pública (CIBERESP), Calle Monforte de Lemos 3-5, 28029 Madrid, Spain; 5Universidad Nacional de Educación a Distancia (UNED), Calle de Bravo Murillo 38, 28015 Madrid, Spain; 6Servicio de Medicina Preventiva, Hospital Universitario 12 de Octubre, Avenida de Córdoba s/n, 28041 Madrid, Spain

**Keywords:** mobility, transmission, networks, infectious diseases, epidemiology, spatial epidemiology, disease modelling

## Abstract

Human mobility drives the geographical diffusion of infectious diseases at different scales, but few studies focus on mobility itself. Using publicly available data from Spain, we define a Mobility Matrix that captures constant flows between provinces by using a distance-like measure of effective distance to build a network model with the 52 provinces and 135 relevant edges. Madrid, Valladolid and Araba/Álaba are the most relevant nodes in terms of degree and strength. The shortest routes (most likely path between two points) between all provinces are calculated. A total of 7 mobility communities were found with a modularity of 63%, and a relationship was established with a cumulative incidence of COVID-19 in 14 days (CI14) during the study period. In conclusion, mobility patterns in Spain are governed by a small number of high-flow connections that remain constant in time and seem unaffected by seasonality or restrictions. Most of the travels happen within communities that do not completely represent political borders, and a wave-like spreading pattern with occasional long-distance jumps (small-world properties) can be identified. This information can be incorporated into preparedness and response plans targeting locations that are at risk of contagion preventively, underscoring the importance of coordination between administrations when addressing health emergencies.

## 1. Introduction

The COVID-19 disease took only a few months from its emergence in the Chinese province of Wuhan to become a global pandemic affecting the world [1]. As with other respiratory viruses, its main route of infection is direct person-to-person contact between an infectious host and a susceptible individual, and therefore, COVID-19 dynamics are shaped mainly by the way both individuals and populations interact with each other [2]. This implies a multi-scale nature of transmission dynamics in which human mobility plays a key role explaining local outbreaks and epidemics [3], geographical diffusion [4,5] and international spread of pandemics [6,7]. 

We have seen a surge in mobility information availability mainly due to the innovative development of big data techniques [8,9,10,11] that the COVID-19 pandemic catalyzed. Since the early stages of the pandemic, numerous studies have been conducted in countries such as China [12,13,14], Italy [15,16], Spain [17] or the USA [5,9,18], among others [19]. Mobility data have been used to model geographical diffusion [12,13,14,15,17,18,19,20], epidemic growth [5,13,16] or to evaluate non-pharmaceutical interventions in place [5,12], primarily by examining the relationship between epidemiological outcomes and changes in the number of travelers from a particular source location. 

Gibbs et al., through exploring mobility trends in mainland China in early 2020, noted that lockdowns were effective in reducing the number of people traveling, but not in changing their destinations, despite restrictions being in place [14]. Mobility here can be understood as a subjacent matrix of connections or a network—a notion long used in epidemic models with coupled metapopulations interchanging individuals (travelers) [21]. Analyzing the structure of mobility networks themselves, rather than considering raw flows as explaining covariates, may provide useful insights on ‘why’ or ‘how’ some phenomena have happened, rather than just describing ‘what’ happened.

Network science methods focus on the study of relationships between elements [22]; the diffusion of diseases in contact or travel networks has been a cornerstone of this field since its inception [22,23]. Two central components can be identified in this context: graph theory, which studies the properties derived from the network's structure and how it determines aspects of disease dynamics [24], and social network, which provides analytical tools for quantifying social behavior (i.e., contact patterns) and relates it to individual and population metrics of centrality, transitivity or modularity; these are useful for statistical and epidemiological models [25,26] that later translate into epidemiological risks, e.g., of infectivity, exposure or transmissibility [22,26,27]. 

This relationship between network methods, disease transmission and the modelling of large-scale geographical mobility networks is an area that requires further investigation to develop evidence-based frameworks in infectious disease epidemiology [26]. Mobility can introduce a new level of complexity with a multiplicity of possible routes, distances and geographical patterns for disease diffusion [7,22]. Such insights can inform public health authorities for future health emergencies if empirical data from actual case studies and large mobility datasets become available to researchers, something that was not accessible before the COVID-19 pandemic.

### Objectives

With this goal in mind, the objective of the present study was to analyze and characterize mobility patterns in Spain, using data gathered by the Spanish Government during 2020–21 [28]. First, raw mobility between the 52 Spanish provinces was turned into a stable matrix of connections, the Mobility Matrix (MM). Second, a directed network was built from the MM after turning raw mobility into a unit of disease transmissibility, the effective distance [7], which enabled us to directly describe both the network and node properties in terms of their epidemiological risk of disease transmission. Third, we used COVID-19 data from the same time period [29] to test how mobility communities within the network coupled the province’s epidemics, thus linking disease dynamics with mobility structure.

## 2. Materials and Methods

### 2.1. Data Sources

Mobility data from 14 February 2020 to 9 May 2021 was gathered by the Ministry of Transportation, Mobility and Urban Agenda (MITMA) while state-of-alarm laws to answer the COVID-19 pandemic were in force [28]. We defined Spain´s 52 provinces as the area of study (Figure 1) and data were aggregated in the form of daily unique travels be-tween all provinces. The COVID-19 data for Spain were available from the Instituto de Salud Carlos III public information dashboard [29]. This information was gathered through the National Surveillance Network (RENAVE, in Spanish). Demographic information was provided by the Instituto Nacional de Estadística (INE) [30].

### 2.2. Mobility Matrix of Spain and Shortest Distance Maps

A description of mobility patterns in Spain was first carried out to characterize behavioral, seasonal and geographical properties. Let $F_{nm}$ be the flux of daily total travelers emanating from a province *m* to any different province *n*, and $P_{nm}$ the fraction of people leaving *m* and going to *n* among all people leaving *m* [7]: (1)$$P_{nm}=\frac{F_{nm}}{\sum_{n\neq m}F_{nm}}$$

The $P_{nm}$ fractions represent the distribution of the relative importance of each destination province *n* connected with *m*. We defined an adjacency matrix containing all $P_{nm}$ pairs for the 52 provinces and named it our Mobility Matrix (MM). We compared a mean MM for the whole 2020 period with subsegments in different 2020 periods by the means of MM ratios and standard deviations (Appendix A) to conclude that there were no significant differences in time, season or measures in place between any of them, thus granting stability. To further study transmission phenomena within this MM, all percentages $P_{nm}$ were turned into a distance-like unit: Effective Distance (ED), as proposed and tested by [7]:(2)$$d_{nm}=1-log\left(P_{nm}\right)$$

ED is a measure of proximity: the larger $P_{nm}$, the lesser $d_{nm}$, and therefore, the closer node *n* is from *m* in terms of disease diffusion [7]. Complex connection patterns emerge from the matrix $P_{nm}$ and its $d_{nm}$ derivate, since EDs allow to mathematically compute paths along the MM that imply multi-step routes due to the aggregative property of logarithms compared with the conditional probability of percentages [7]. We can find the shortest distances between two provinces $D_{nm}$ as the minimal sum of $d_{nm}$ required to reach node *n* from *m*.

For example, if $d_{BA}=6$ but $d_{CA}=2$ and $d_{BC}=2$, any travelling disease would arrive from A to B faster by transitioning through C $\left(D_{BA}=4\right)$ rather than directly from A to B. The set of shortest paths between pairs of provinces within the MM was obtained using the classical Dijkstra’s algorithm [31] implemented in the *igraph* R package [32]. For each province, a map of all the shortest incoming and outgoing $D_{nm}$ was created.

An MM built from effective distances informed of the relative connectivity strengths of the nodes of a network in relation to each node's mobility flow, rather than the raw numbers of moving individuals. As shown in [7], this quantity was empirically proven to better reflect the role of human mobility with the spread of infectious diseases than the untransformed flows. Including effective distances in a network is a novel approach that aims to improve the generalizability of our results to epidemiological conclusion.

### 2.3. Epidemic Diffusion Network and Measures

From the MM, we defined a network [33] to represent and uncover these complex mobility patterns that might drive diffusion events. To model the network, only relevant connections (e.g., those with $P_{nm}$ over 10%, see Appendix A) were selected. When applied to the study of disease diffusion dynamics, we named this network the “*Epidemic Diffusion Network*” (EDN) and studied several network measures with epidemiological relevance to the spread of diseases (Appendix A).

In a network, the degree of a node is the number of edges connected to it (incoming and outgoing)—a representation of connectiveness. In weighted networks, the strength of a node is the sum of the weights of each edge connected to the node, i.e., the weighted degree. Degree and strength are local measures that account only for the neighboring properties of nodes. Betweenness, on the other hand, sums up the number of times the node lies within the path between any other two nodes, thus scoring nodes that act as a bridge or intersecting points [27,34]. The three are estimated for each node of the EDN.

Network substructure analysis was performed to detect communities using the Walktrap [35] and Infomap [36] algorithms that were implemented in the igraph R package [32]. These were chosen based on the “Question Alignment” method, considering the nature of the research question, given that Walktrap and Infomap are better suited for community detection for transmission events between nodes [37]. A found community must be understood as a mobility cluster in epidemiological terms. This implies that nodes are connected in such a way that movement (flow) within the community is much more likely. The percentage of total movements happening within communities rather than between them is known as the modularity score. The clustering of nodes happens hierarchically and will be represented with a dendrogram.

### 2.4. Network Structure and COVID-19 Dynamics

Since mobility drives the geographical diffusion of COVID-19, it would be expected that provinces within the same community, which share the most of their mobility fluxes, also share more similar disease dynamics due to increased multi-seeding processes [17]. These dynamics are summed up in the cumulative incidence in 14-day (CI14) epidemic curves; therefore, we can compare their trajectories, shapes and magnitude throughout the study period.

We adopted an analytical approximation by proposing a measure of curve proximity. We can compute the absolute difference in daily CI14 between any two provinces m,n for a given day *t*: (3)$$d_{mn,t}=\left|\mathrm{CI}14_{m,t}-\mathrm{CI}14_{n,t}\right|,$$
where $\mathrm{CI}14_{m,t}$ denotes the CI14 at province *m* on day *t*. The mean $\bar{d_{mn}}$ of these daily differences over the time under study is the global proximity measure between the disease dynamics of any two provinces. The lower $\bar{d_{mn}}$ value, the more similar COVID-19 dynamics that two provinces have experienced.

The 52 × 52 possible $\bar{d_{mn}}$ are calculated between each province and (1) the rest of the provinces within its community and (2) the rest of the provinces outside. All obtained means are compared with a T test with the null hypothesis of no differences in $\bar{d_{mn}}$, i.e., being part of a community does not imply a difference in disease dynamic for the studied period.

All analyses were made using R version 4.0.3 [38] and the network package igraph [32].

## 3. Results

### 3.1. Mobility Matrix of Spain

The daily aggregated total mobility from the 14 February 2020 to 9 May 2021 is shown in Figure 2. The first week (14–21 February 2020) is considered the reference mobility value before the pandemic began. The first sharp decline in March 2020 happens after the first declaration of the stay-at-home order, which was lifted in June when mobility almost reached pre-pandemic values. By the end of August, it declines again and does not recover until the end of our study period, even after the second state of alarm came into force from 24 October to 9 May.

Using the example of the relationship between two provinces, we see how raw $F_{nm}$ give place to the stable patterns of provinces’ connections: originally, the flux between them is identical in both ways (Figure 3A) and follows the variations depicted in Figure 2. However, once turned into the $P_{nm}$, we learn that the originally identical $F_{nm}$ now represent very different percentages for each of them: ~80% of total travels for one and only ~25% for the other (Figure 3B). $P_{nm}$ are stable connection patterns that are not affected by seasonality or measures in place and give birth to each province’s set of connections (Figure 3C,D), where major fluxes that will conform the EDN can be clearly discerned from the larger number of less relevant connections existing. A deeper explanation of this rationale with examples can be found in the Appendix A.

### 3.2. Epidemic Diffusion Network and Shortest Distance Maps

The 52 provinces are the network´s nodes and, between them, 135 weighted edges [33] represent the main MM connections, as defined in the Appendix A. An example of the shortest distance maps is shown in Figure 4. The modelled network is presented in Figure 5A. Madrid scores the highest degree (11), followed by Valladolid (9), Araba/Álava (8), Málaga (8), Sevilla (8) and Valencia (8). Madrid, Valladolid and Araba/Álava also record the highest strength (25.7, 25.3 and 20.2). The network flow betweenness, on the other hand, concentrates in provinces of the southernmost community. The mean (SD) values for the entire network are 5.2 (2) degrees, 12.5 (5.5) strength and 318.1 (408.9) flow betweenness (Appendix A).

A total of seven communities are detected within the network, not differing between Walktrap and Infomap algorithms (Figure 5 and Appendix A). The biggest community is the southernmost, consisting in a total of twelve provinces and connected to the outside only by two provinces. On the other hand, the smallest communities are the two composed by the Canary Islands (two provinces) and Galicia (four provinces). The first is dependent only on Madrid as the main connection. The other four communities represent different geographical regions. The highest scoring provinces in terms of degree/strength are well divided among communities. The network´s modularity is 63%. The hierarchical clustering dendrogram and qualitative conclusions derived from it are depicted in the Appendix A.

### 3.3. Network Structure and COVID-19 Dynamics

COVID-19 CI14 epidemic curves by provinces grouped by the community can be seen in Figure 6. Provinces in the same network community seem to share a more similar COVID-19 incidence, represented by their curves’ trajectories, shapes and magnitude. This is confirmed by the fact that all provinces have a statistically significant lower mean daily difference $\bar{d_{ij}}$ with other provinces within their same community compared with the outer ones (Figure 7).

Three communities show a smaller $\bar{d_{ij}}$ between inner and outer provinces: Central–East (138.4 vs. 150.5), Northeast (131.1 vs. 143) and Central–West (101.7 vs. 133.7) communities. They encompass 28 out of 52 provinces together, including the main cities such as Madrid, Barcelona, Valencia, Valladolid or Zaragoza, with all having experienced a significant epidemic burden (Figure 6). They are followed by the South (101.7 vs. 133.7) and North (111 vs. 152.9) communities, which showed greater mean CI14 differences with the outside provinces. Finally, the greatest differences are found for the Canary Islands (49.9 vs. 175.3) and Northwest (50.3 vs. 130) communities, noting, at the same time, that these two communities were the least impacted in terms of CI14 (Figure 6), and both remain to be peripheral small communities (two and four provinces) within the network (Figure 5).

## 4. Discussion

We used raw mobility data gathered during the COVID-19 pandemic to curate and develop a Mobility Matrix (MM) in the form of an adjacency matrix between the 52 provinces of Spain; this was used to build a network model to unveil complex properties that drive geographical diffusion phenomena linked to mobility (such as, but not limited to, epidemics). This network has been described, together with its properties, and linked to COVID-19 dynamics between March 2020 and May 2021.

Mobility has been analyzed and described in successive layers of complexity: first, by depicting the raw fluxes of daily travelers. Secondly, by turning them into a distance-like variable and unveiling the constant relationships and patterns emerging from the data. Third, by building up a network model to further understand diffusion phenomena within that network. The latter was mainly achieved through hierarchical clustering for community detection and the shortest distance maps. Each of these layers of study allowed for us to gain a share of insight into the complexity of human mobility in one specific way.

### 4.1. Mobility Matrix(MM) and its Derived Effective Distance Network (EDN)

The MM reveals that diffusion phenomena are governed by the provinces’ set of major bilateral fluxes that remain constant in time despite natural variations such as weekends, holidays or seasonal (i.e., summer) mobility. Even external variations such as mobility restrictions that were put in place were able to reduce the number of people travelling (Figure 2) but not their usual destinations (Figure 3). Each province has a unique set of main neighbors and ways of interacting with them. These constitute the properties of provinces which will determine their role in diffusion phenomena as the nodes of a network, connected by their relevant effective distances on the edges: the EDN. 

The EDN shows a configuration of hubs that connect with smaller provinces rather than between them, acting more like local gravitating poles for adjacent lesser provinces rather than drivers of the whole network (Figure 5). This is the case for the four largest degree-scoring provinces: Madrid, Valladolid, Araba/Álava and Málaga, each belonging to one big community (Appendix A). They are followed by other important provinces of the same communities (Sevilla, Valencia…). However, the other big community, Northeast, does not appear until the 12th and 13th position with Zaragoza and Barcelona, the latter being the 2nd largest city of Spain. Contrary to the rest of communities, the Northeast shows a completely different behavior in which no main hub emerges. We are unable to explain these findings and their implications in diffusion dynamics, but we hypothesize that this subnetwork would be capable of self-sustaining a disease loop more effectively; we hypothesize this due to positive infective feedback emanating from the horizontal structure of shared connections of equal importance, with no “secondary” provinces just sending a large flux to the hub; however, a disease-modelling-in-networks approach would be required to confirm this hypothesis.

Out of the seven found communities, only the smallest two (Canary Islands and Northwest) are composed of a single full administrative region while the rest bring together provinces under different administrations at the level of Autonomous Communities—an outcome that similar studies also found when assessing mobility clusters [39]. This is especially true for the second and third largest central communities, with Madrid acting as a main bridge between two mobility clusters and several Autonomous Communities (including the Canary Islands). Hierarchical clustering (Appendix A) shows how the southern community conforms an independent area from the rest of Spain. This also reflects the fact that these provinces score the highest in flow betweenness (Appendix A), since this is a metric of global transit, and 12 provinces are connected by only two bridges, thus tarnishing this measure in our network. 

A relationship between the EDN-found communities and CI14 evolution at the province level in Spain exists (Figure 7), linking mobility patterns and spatial diffusion of a transmissible disease in a time period when public health measures targeted mobility between provinces. Given that the majority of mobility communities share more than one Autonomous Community, these findings stress the importance of coordinated, data-driven and locally wise measures.

### 4.2. Network Epidemiology

The network on which a disease spreads highly determines the evolution and dynamics it will experience [22]. We observed how several studies used mobility data to study the early stages of the COVID-19 diffusion [12,13,14,15,16,17,18,19,20] but did not address the subjacent properties of the countries’ mobility networks. By refining mobility data from Spain into a new approach within network modelling, we enabled social network analysis at a large geographical scale, treating each province like an individual relating with its neighbors. We were then able to establish some parallelisms with previous network studies.

Centrality measures such as degree, strength or betweenness have been investigated in COVID-19 contact tracing data. Two examples from India and South Korea [40,41] found that out-degree could identify a small proportion of patients responsible for a large portion of secondary infections. In addition, individuals with high betweenness scores acted as ‘bridges’ between subcomponents in the data [41]. Contact-tracing is always a retrospective approach, but if the mobility structure is stable, we can identify the most relevant locations before any outbreak and prepare for it in advance. Degree and strength identify well-connected individuals that are, at the same time, more exposed to a disease and at a higher risk of transmitting the disease onward, while betweenness identifies bridging locations that can be targeted to geographically contain an infection. All these measures are highly context-dependent [27,42], acquiring different meanings in different kinds of networks, especially where substructure is a relevant network component, such as the EDN [25].

This structure links individual behavior to broad population effects in networks. For example, during the 2014–2016 Sierra Leona Ebola outbreak, changes in the pattern of the case-contacts network forecasted the final epidemic size [43]. During the COVID-19 pandemics, governments implemented measures aiming to alter the social network of contacts by social distancing, travel bans or mandatory mask use. Their effects can be measured in terms of network measures [40] or they can be modelled as intervention scenarios for pandemic control [44].

The network structure has a significant impact on the analysis and interpretation of the entire network [25]. At the population (spatial) level, a mobility network such as the EDN turns an individual set of contacts into a series of infective distances, shortest paths and subcommunities [22]. These can be compared with several idealized types of networks found in natural, social or technological systems [45], among which small-world networks are one of the most relevant [46]. They are characterized by a high clustering degree and short path lengths, similar to those found in the EDN, but they also have random links that connect otherwise distant elements of the network.

Diseases that spread on small-world networks take the form of a spreading wave from an origin source, with the ability to quickly reach distant elements [22]. This was observed in the international air-travel network during the pandemics of H1N1 in 2009 or SARS in 2003, as described by Brockman and Helbing [7]. A similar pattern was seen in the diffusion of COVID-19 during the summer of 2020 in Spain, where it started in a region (northeast Spain) and then quickly spread to Madrid, from where it reached the rest of Spain [47], a path depicted by the effective distance map route in Figure 4B.

High modularity was also found in our network which, according to some authors, means that the disease is more likely to remain trapped within communities [48]. However, other authors consider this effect to be modest, especially in highly contagious diseases which would only be trapped with a modularity of over 80%, which is rare [49]. Despite this, modularity is still an essential factor to consider for control strategies in epidemiology. In the presence of a less transmissible disease, appropriate measures can increase the "structural trapping" of the network, which refers to the ability to prevent the disease from spreading from one community to another [49].

### 4.3. Limitations

The main limitation of the study comes from the MM and EDN design, which was based on the researchers’ decisions. Should those decisions have been different, the network structure, measures or communities—or its interpretation—might differ [22,24]. When working with networks, there is no common framework for deciding which structure or measures are the most useful [26], with each network yielding different measures’ meaning in its specific context [27]. The same applies to community detection algorithms, as previously stated [37]. 

The epidemiological implications of our findings in the geographical scale comes from research in social network analysis in individual groups of people. By turning Brockmann and Helbing’s effective distance [7]—an epidemiological measure of disease spreading through mobility—to the network’s edges, we can link the individual and spatial domains. However, more recent developments of the effective distance metric are available that better approximate reality, going beyond the mere shortest paths through multiple random walks at the same time [50]. We decided to continue with the first version, since modelling more complex epidemic dynamics would require much more than only accounting for multi-seeding processes, and we focused on the mobility network itself rather than the disease network.

The chosen administrative unit for this work was the province since public health units usually operate at this level. However, this approach might hide subtle mobility patterns at a smaller level, e.g., municipalities, which could lead to different conclusions or otherwise would be needed when dealing with more localized problems.

## 5. Conclusions and Recommendations

It is only by addressing together information from network measures, structure and modularity that a complete picture can be drawn for the epidemiological implications of the MM and the EDN in Spain. An analysis of the public health implications should encompass three aspects: (1) the community of the origin province, (2) the nearest destinations and the effective distance maps and (3) the risk of spreading to the entire network.

For example, an outbreak starting in the South community is more likely to remain trapped since only two bridging provinces connect it with the rest of Spain (Figure 5), and therefore, early measures should focus on those provinces to prevent the disease from spreading throughout the country. Similarly, if the source of the outbreak is a small province linked to a main hub (as shown in Figure 3D), a strategy could be to target directly third-order provinces linked to that hub, since the spreading from the hub is much more unlikely to be prevented. Madrid acts as the main bridge between most of the mobility communities, reflecting the importance of the capital city and the socio-economic distributions of travel in Spain, playing a central role in disease diffusion in the country. However, this hub effect is less relevant on the second-largest regions of Spain, Barcelona and its mobility community of the Northeast where no main hub emerges and all provinces relate equally to each other. All these epidemiological correlates could be incorporated into preparedness and response plans at the subnational, national and international levels, since the method is scalable to different levels of aggregation and disease dynamics are multi-layered in nature.

Further research is needed to test the relations between network components and disease epidemiology. Simple mathematical (e.g., SEIR) models are able to fully capture the complexity of these dynamics when paired with the adequate subjacent network [3]. The massive amount of epidemiological and social information made available during the COVID-19 pandemic poses a unique opportunity to address the identified gaps in knowledge and help us advance towards a more generalized theoretical framework in the field.

## Figures and Tables

**Figure 1 ijerph-20-04356-f001:**
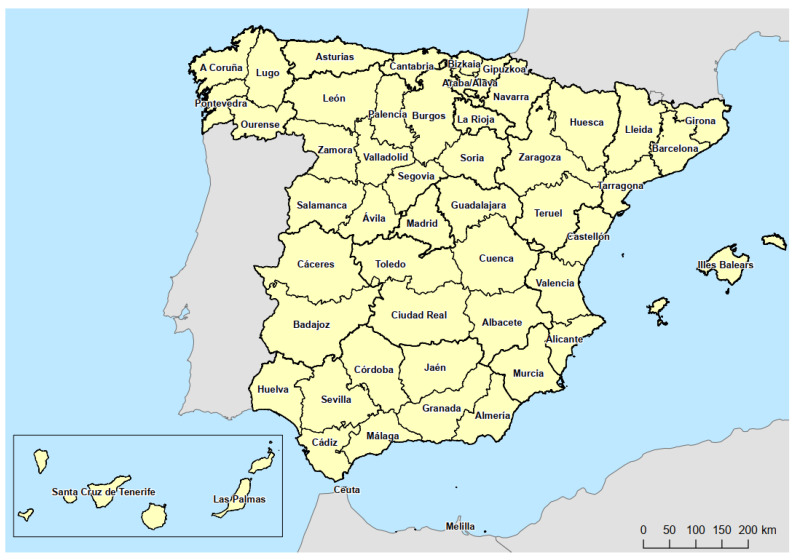
Political map of Spain.

**Figure 2 ijerph-20-04356-f002:**
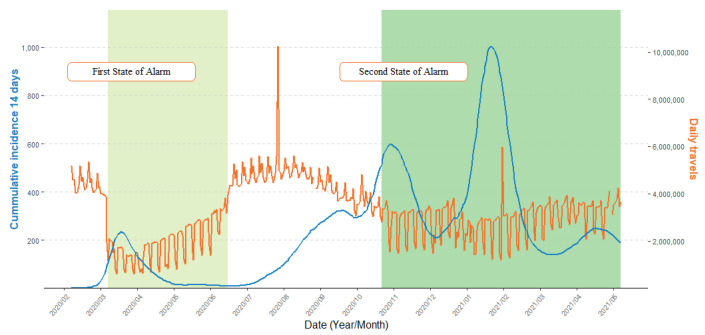
Daily total travels between provinces and the 14-day cumulative incidence in Spain, 14 February 2020–9 May 2021. Cumulative incidence in 14 days per 100,000 inhabitants.

**Figure 3 ijerph-20-04356-f003:**
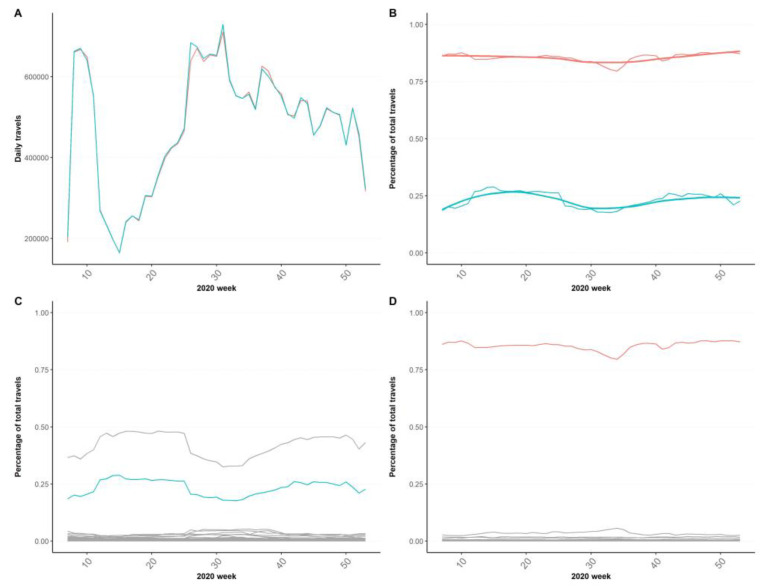
Rationale for defining the Mobility Matrix of Spain from unique daily travelers. Red line: flux from Guadalajara to Madrid; Blue line: flux from Madrid to Guadalajara. (**A**) Total unique travelers (*F_nm_*) by 2020 week between Madrid and Guadalajara and (**B**) Percentage of total travelers (*P_nm_*) by 2020 week between the same provinces. (**C**) Set of percentages of total travels from Madrid to all other provinces it sends travelers to. (**D**) Set of percentages of total travels from Guadalajara to all other provinces it sends travelers to.

**Figure 4 ijerph-20-04356-f004:**
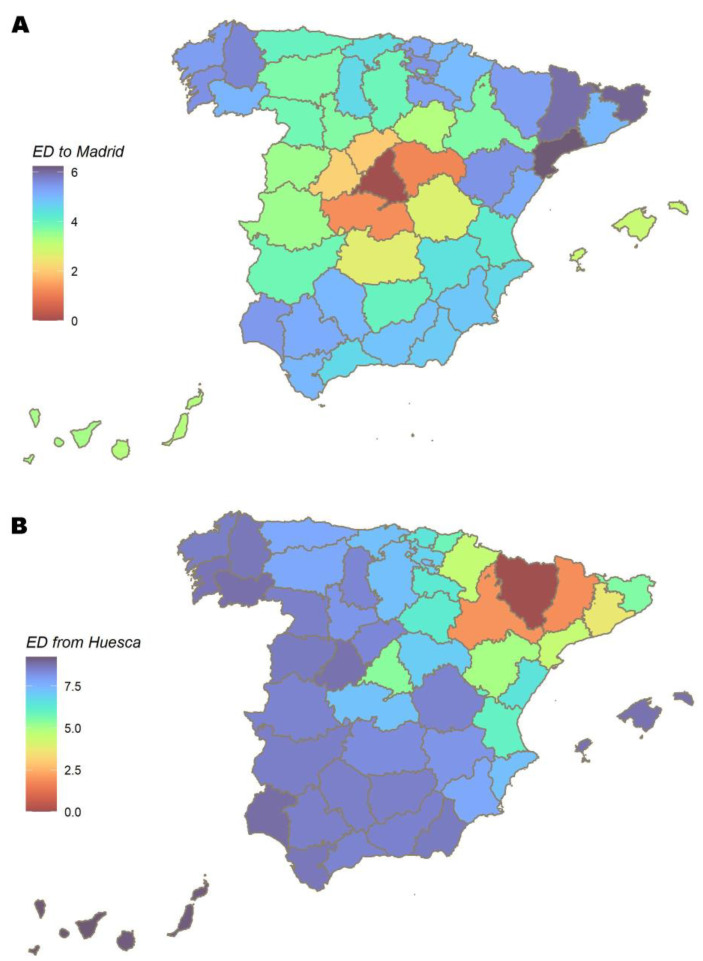
Shortest distance maps from (outgoing) and to (incoming) two provinces, Huesca and Madrid. (**A**) Shortest distances to Madrid. (**B**) Shortest distances from Huesca.

**Figure 5 ijerph-20-04356-f005:**
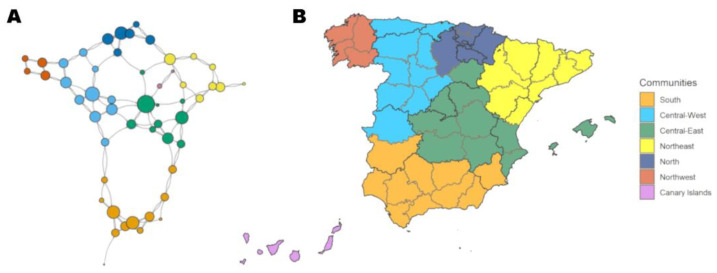
Effective distance network (EDN) representation of network measures and map of communities. (**A**) Vertex size is proportional to incoming/outgoing connections (degree). Each edge represents a connection. Communities are represented by color. (**B**) A map of Spain with provinces colored by community belonging.

**Figure 6 ijerph-20-04356-f006:**
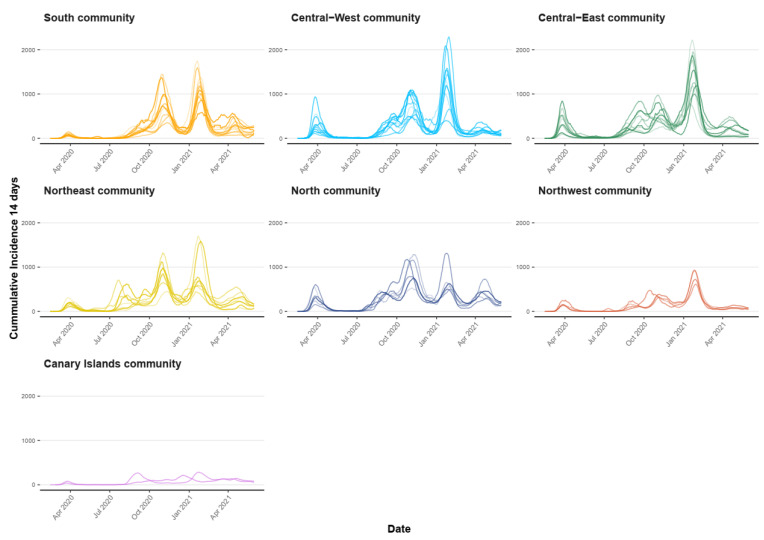
CI14 per 100,000 inhabitants by province, grouped by community during the study period.

**Figure 7 ijerph-20-04356-f007:**
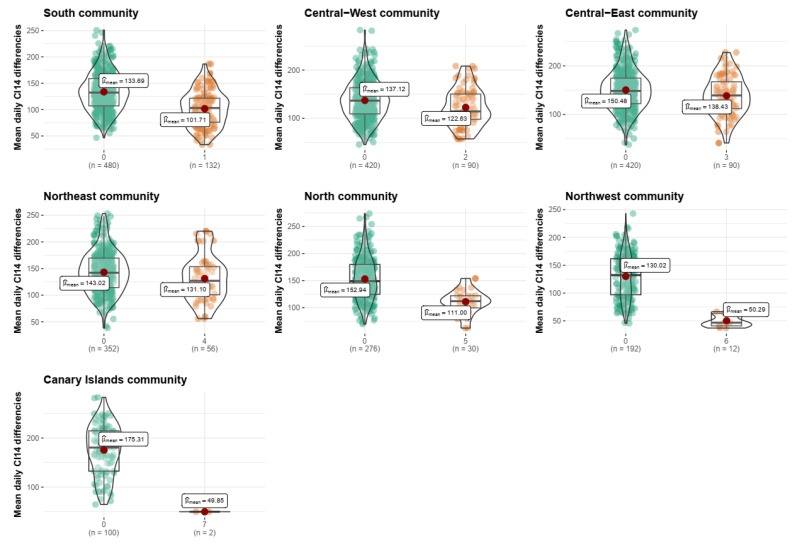
Mean daily differences between CI14 curves within communities’ vs. outside community for every detected community. All mean differences between groups for all communities were statistically significant. X axis: ‘0’ stands for all the provinces outside the represented community.

## Data Availability

The datasets used and/or analyzed during the current study are publicly available from the following sources: Instituto de Salud Carlos III. Situación y evolución de la pandemia de COVID-19 en España. Available from: URL: https://cnecovid.isciii.es/covid19/ (accessed on 22 February 2023). Ministerio de Transportes, Movilidad y Agenda Urbana. Estudio de movilidad con Big Data. Available from: URL: https://www.mitma.gob.es/ministerio/covid-19/evolucion-movilidad-big-data (accessed on 22 February 2023).

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
