# Peer review of "Epidemic Diffusion Network of Spain: A Mobility Model to Characterize the Transmission Routes of Disease"

_ijerph, 2023, doi:10.3390/ijerph20054356_

Round 1
Reviewer 1 Report
This paper studied the mobility in diffusion of disease at different scales.
This paper is well-written in general. The topic of disease transmission routes is important and intersting to readers. The methodology is sound and the results are promising.
Previous studies on epidemic growth, travel networks and mobility could be further discussed. Instead of just mentioning those works, a more detailed comparison covering their contributions and drawbacks could be further provided.
Although the study is focused on Spain, the methodology should be further discussed in regard of the generalization ability.
In section 2.2 and 2.3 detailed network structures and measures are introduced. A depiction of the structure and possible table summaries of the measures are suggested to add to the revision for easier understanding of the novelty of the proposed network.
Figure 2 should be high resolution, the quality of the figure should be improved. What does the different color means in the figure? In the figure caption, only blue and red are mentioned. I think a separate paragraph explaining with more details and some related legends on the figure are necessary.
The intuition behind the novel design of mobility matrix coud be further emphasized.
Author Response
Reviewer 1
- This paper studied the mobility in diffusion of disease at different scales. This paper is well-written in general. The topic of disease transmission routes is important and interesting to readers. The methodology is sound and the results are promising.
We would like to thank reviewer 1 for the interesting comments and for valuing our work. We hope we have address all his/her concerns to the best of our extent, all of which have improved the paper overall.
- Previous studies on epidemic growth, travel networks and mobility could be further discussed. Instead of just mentioning those works, a more detailed comparison covering their contributions and drawbacks could be further provided.
We have extended the introduction section dedicated to previous mobility studies. We have made explicit claims of how previous methodologies lacked the insight that we are addressing in our paper, and why we believe our findings are an innovative way of using the newly available amount of data generated during the pandemic period.
Also, since another reviewer pointed to similar comments, we have also extended the discussion section related to mobility in networks and their importance in epidemiology of infectious diseases.
- Although the study is focused on Spain, the methodology should be further discussed in regard of the generalization ability.
We have added another discussion section, “Final considerations”, that discuss the public health implications of our findings. At the end of the second paragraph, we explain how the same conclusions reached as the provinces levels of Spain could be escalated to international levels or, on the contrary, to subnational levels if we wants to study specific regions. The interpretation remains the same, as detailed in the whole paragraph, as long as mobility information is available and can be processed in the same way.
- In section 2.2 and 2.3 detailed network structures and measures are introduced. A depiction of the structure and possible table summaries of the measures are suggested to add to the revision for easier understanding of the novelty of the proposed network.
We have added a table in the supplementary materials (Table S2), that sums up the relevant aspect of those measures and their relation to epidemiological diffusion parameters of interest. In the main text, it is references just at the beginning of section 2.3 in the first paragraph
- Figure 2 should be high resolution; the quality of the figure should be improved. What does the different color means in the figure? In the figure caption, only blue and red are mentioned. I think a separate paragraph explaining with more details and some related legends on the figure are necessary.
We have made changes in the figure highlighting the same colour code in all 4 panels, which greatly improves the understanding of the figure. Now only the two presented flows are visible an can be understood together with the explanation in the paragraph above the figure. Figure caption has also been improved for adding clarity.
- The intuition behind the novel design of mobility matrix coud be further emphasized.
We have added a final paragraph to section 2.2: “A MM built from effective distances informs of the relative connectivity strengths of the nodes of a network in relation to each node's mobility flow, rather than the raw numbers of moving individuals. As shown in [7], this quantity has been empirically proved to better reflect the role of human mobility with the spread of infectious diseases than the untransformed flows, and including effective distances in a network is a novel approach that aims to improve the generalizability of our results to epidemiological conclusion.”

Reviewer 2 Report
Thank you for an opportunity to review your manuscript.
All parts of the manuscript are well written and well structured. I found no major flaws. The main topic of the article is well explained, therefore your model could be helpful and adopted by various European public health administrations. In line 368, please change the word "don't" with "do not".Author Response
Thank you for an opportunity to review your manuscript. All parts of the manuscript are well written and well structured. I found no major flaws. The main topic of the article is well explained; therefore, your model could be helpful and adopted by various European public health administrations.
We would like to extend a warm thank you to Reviewer 2 for his/her kind words. It is an honor to receive such a high degree of recognition from a peer
- In line 368, please change the word "don't" with "do not".
changed
Reviewer 3 Report
The authors have chosen a topic that deserves timely intervention and discussion.
Suggestions:
Strengthen theoretical background. Connect theory with discussion. Introduce underlying theories in the literature review. Elaborate in more depth.
List the objectives in a separate section.
Literature review: Elaborate on the research gaps. How will it contribute to the existing literature?
Include a separate research gap section.
Line 79: Is the data sufficient? What would be the rationale behind analyzing 15 months of data?
Discussion: Should echo with literature review section; Elaborate on implications and future research directions in more depth.
Line 277: “. We are unable to explain these findings, but the resulting sub-network points towards a differentiated diffusion dynamic compared with the rest of Spain.” -> what would be a possible explanation? How would your findings compare to those pertaining to similar studies in the literature?
Provide concrete examples or strategies for the implications section, e.g., in the abstract: “This information can be incorporated to preparedness and response plans targeting locations that are at risk of contagion preventively, underscoring the importance of coordination between administrations when address- health emergencies.” -> which locations should be targeted?
Elaborate on the limitations section in more depth and clarity.
Author Response
Reviewer 3
We want to express our gratitude toward all the comments made, that have improved the quality of the paper. Based on them, we made extensive improvements in the introduction and discussion section to make more explicit claims about our approach to the matter, the literature background and current gaps in the evidence that we are covering.
We would like to address together all these comments made by Reviewer 3:
- The authors have chosen a topic that deserves timely intervention and discussion. Strengthen theoretical background. Connect theory with discussion. Introduce underlying theories in the literature review. Elaborate in more depth.
- Literature review: Elaborate on the research gaps. How will it contribute to the existing literature?
- Discussion: Should echo with literature review section; Elaborate on implications and future research directions in more depth.
- Provide concrete examples or strategies for the implications section, e.g., in the abstract: “This information can be incorporated to preparedness and response plans targeting locations that are at risk of contagion preventively, underscoring the importance of coordination between administrations when address- health emergencies.” -> which locations should be targeted?
In the introduction, we now cover how studies using mobility are missing a very important aspect of mobility information that, for the first time, we had available for tackling the COVID-19 pandemic. More that discussing their findings, which are useful and was valuable at that time, we point the fact that a deeper insight is going missing, with the chance to interrogate mobility to better understand its underlying social structure and how that conditions the spread of diseases, and why network methods are the most appropriate for answering those kinds of questions. A final explicit section for objectives has been added, as suggested, while for the gap section we make an explicit paragraph just before objectives.
Later, in the second section of the discussion, we address how our findings can compare with previous knowledge presented in the introduction. The measured metrics are discussed in context from an epidemiology point of view.
Finally, a new section called “final considerations” has been created to sum up the outcomes of our finding for epidemiology, by providing several examples based on the landscape described in the rest of the paper of how public health authorities could use this information and how future research is needed to consolidate and validate this approach.
Taken together, we hope that the modifications included in the new version of the manuscript cover the full extent of concerns expressed by Reviewer 3.
- Line 79: Is the data sufficient? What would be the rationale behind analyzing 15 months of data?
Mobility data was gathered by public authorities in Spain thanks to special powers granted by the state of alarms laws passed by the Spanish Parliament. Even though it was in place for two different periods (March 14th – June 21st) and October 25th – May 9th, mobility data kept being collected to inform the evolution of public health travels ban in place. That is why we use 15 months of data, and why we developed a working framework for extracting constant patterns of mobility, knowing that the information flow would be over at some point and wanting to maximize its impact and future usability.
- Line 277: “. We are unable to explain these findings, but the resulting sub-network points towards a differentiated diffusion dynamic compared with the rest of Spain.” -> what would be a possible explanation? How would your findings compare to those pertaining to similar studies in the literature?
We have hypothesis regarding this fact; however, we are not able to prove them with our methodology nor comparing with literature since our approach is novel in this context. We have added a brief explanation in the same paragraph:
We are unable to explain these findings and their implications in diffusion dynamics, but we hypothesize that this subnetwork would capable of self-sustaining a disease loop more effectively thanks to positive infective feedback emanating from the horizontal structure of shared connections of equal importance, with no “secondary” provinces just sending a large flux to the hub, but a disease modelling in networks approach would be required to confirm this hypothesis.
- Elaborate on the limitations section in more depth and clarity.
We have re-elaborated this section, together with the other changes. We believe it is now clearer, more coherent and several redundancies were eliminated to grant clarity and concision.
